# Recent Advances in Tomato Gene Editing

**DOI:** 10.3390/ijms25052606

**Published:** 2024-02-23

**Authors:** Eduardo Larriba, Olha Yaroshko, José Manuel Pérez-Pérez

**Affiliations:** Instituto de Bioingeniería, Universidad Miguel Hernández, 03202 Elche, Spain; oyaroshko@umh.es

**Keywords:** zinc finger nucleases, TALE nucleases, CRISPR/Cas, *Solanum lycopersicum*, tomato domestication, plant architecture, abiotic stress, pathogen resistance

## Abstract

The use of gene-editing tools, such as zinc finger nucleases, TALEN, and CRISPR/Cas, allows for the modification of physiological, morphological, and other characteristics in a wide range of crops to mitigate the negative effects of stress caused by anthropogenic climate change or biotic stresses. Importantly, these tools have the potential to improve crop resilience and increase yields in response to challenging environmental conditions. This review provides an overview of gene-editing techniques used in plants, focusing on the cultivated tomatoes. Several dozen genes that have been successfully edited with the CRISPR/Cas system were selected for inclusion to illustrate the possibilities of this technology in improving fruit yield and quality, tolerance to pathogens, or responses to drought and soil salinity, among other factors. Examples are also given of how the domestication of wild species can be accelerated using CRISPR/Cas to generate new crops that are better adapted to the new climatic situation or suited to use in indoor agriculture.

## 1. Introduction

Current global agricultural production faces significant losses due to increased environmental stress caused by climate change. This stress results from the fluctuations in temperature and precipitation, along with related factors such as soil salinization [1,2]. Consequently, the development of new tools to alleviate the negative effects of the current climate change scenario is crucial. In one avenue of development, to effectively respond to the various stresses caused by anthropogenic climate change, crops are currently undergoing genetic modification through new genomic techniques (NGTs) to enhance their resilience and adaptability [3]. Gene-editing tools, such as zinc finger nucleases (ZFNs), transcription activator-like effector nucleases (TALENs), and clustered regularly interspaced short palindromic repeats (CRISPR)/CRISPR-associated proteins (Cas), are the most commonly used NGTs for crop improvement.

Tomatoes (*Solanum lycopersicum* L.) are the most widely grown vegetable crop, with 186.1 million metric tons produced worldwide in 2022 [4]. However, drought, increasing global temperatures, the spread of insect pests, bacteria, and viruses pose a threat to their current production [5]. To overcome this, gene-editing tools have increased the tolerance of tomato plants to adverse factors, including heat, drought, salinity, bacteria, and viruses. Furthermore, tomato plants with increased levels of lycopene and carotenoids, along with other biochemical enhancements, have been effectively produced using gene-editing tools. Against this background, in this review, we present an overview of the gene-editing tools that are available and the strategies used to obtain edited plants, the results attained in tomatoes when using these tools, and a discussion of the weaknesses and strengths of the current approaches.

## 2. Gene-Editing Systems in Plants

### 2.1. Zinc Finger Nucleases (ZFNs)

Cys2-His2 (C2H2) zinc finger domains are present in many eukaryotic transcription factors. The classical C2H2 domain, with 28–30 amino acids, comprises two β-strands and an α-helix stabilized by a Zn^2+^ atom through conserved cysteine and histidine residues. Each C2H2 domain binds 3–4 nucleotides of DNA. Furthermore, the fusion of tandem C2H2 domains, separated by a conserved sequence of five amino acids, enables the binding to specific, longer DNA sequences. The zinc finger transcription factors were discovered in 1985 [6] and have since been utilized to regulate gene expression. ZFNs are chimeric proteins that comprise multiple C2H2 zinc finger domains, up to three, which are fused to the C-terminal cleavage domain of the endonuclease FokI, rendering ZFNs pioneering gene-editing tools to have been made accessible. The DNA sequence specificity of composite arrays of C2H2 zinc finger domains utilized to construct ZFNs determines the specificity of endonuclease cleavage. Dimerization of two different ZFN proteins results in a double strand break (DSB) in DNA with 5–7 nucleotide 5′ overhangs, as the Fok1 cleavage domain needs dimerization to cleave DNA (Figure 1a) [7,8]. The resulting DSBs in the DNA initiate endogenous mechanisms of DNA repair, predominantly through the non-homologous end-joining (NHEJ) pathway, which leads to the appearance of deletions and insertions at the repair site [9].

The editing efficiency of the ZFN system is generally poor, with a success rate ranging from only 1% to 10% [9,10,11,12,13]. This limitation could be due to potential interactions between the neighboring C2H2 domains in the ZFNs, which affect their DNA binding specificity, although the underlying mechanism remains unclear [14]. Due to their low specificity, ZFNs can cause unintended mutations at off-target sites, as evidenced by earlier studies [8]. The first plant to be edited using the ZFN system was the model plant *Arabidopsis thaliana* [15], followed by maize [9], tobacco [11,16,17], and soybean [18]; however, the scope of crop editing using ZFNs has been limited due to the potential drawbacks of this approach [19,20,21].

### 2.2. Transcriptional Activator-like Effector Nucleases (TALENs)

TALENs are also chimeric proteins that can be engineered to cleave particular DNA sequences. Similar to ZFNs, TALENs comprises the C-terminal domain of FokI endonuclease and specific DNA binding domains, originating from the transcription activator-like effectors (TALE) from plant-pathogenic bacteria [22]. The TALE domain consists of numerous (between 12 and 27) consecutive repeats of 33–35 amino acids, each containing two α-helices and a short repeat variable diresidue known as RVD. RVD is essential for making sequence-specific DNA contacts. By using both bioinformatics tools and empirical data, the association between the RVD amino acid sequence and its binding to a specific nucleotide in the DNA was established [23,24]. Shortly afterwards, newly engineered *TALE* genes were utilized to selectively upregulate endogenous genes in various plant species [25]. The assembly of repeat modules for different RVD arrays allows for the generation of sequence-specific DNA binding domains to whatever length is desired. Like ZFNs, two monomeric TALENs are necessary to bind the target sites in the DNA and enable FokI to dimerize and cut the DNA (Figure 1b). *Arabidopsis thaliana* [26] and rice [27] were the first plant species to be edited using TALENs. To date, over 50 genes in 12 plant species, such as maize, wheat, barley, tobacco, soybean, potato, and tomato, among others, have undergone successful gene editing through the use of TALENs [28].

When considered in comparison to ZFNs, the interactions between DNA-binding domains of TALENs and their target nucleotides are simpler than those between ZFNs and their target trinucleotides. This means TALEN design is generally less complex than that of ZFNs. Additionally, TALENs offer other benefits over ZFNs, including higher editing efficiency (approximately 30%), and lower off-target mutations [28]. However, a clear disadvantage of TALENs is their significantly larger size when compared with ZFNs, which can pose challenges for their delivery and expression. 

### 2.3. Clustered Regularly Interspaced Short Palindromic Repeats (CRISPR) and CRISPR-Associated Protein (Cas)

CRISPR and its associated locus encoding Cas proteins were initially discovered in the genomes of bacteria [29] and archaea [30]. The CRISPR/Cas system plays a crucial role in the adaptive immune responses against lytic bacteriophages and plasmids [31,32,33]. Upon initial infection by a phage, fragments of its DNA integrate into the CRISPR genomic locus. Upon new viral infection, the CRISPR locus is transcribed into the CRISPR RNA (crRNA), consisting of spacers with distinct sequences from various phages, indicative of past infections. These spacers are separated by repetitive endogenous sequences, known as direct repeats [34]. The Cas proteins possess nuclease and helicase domains that facilitate identifying and cutting exogenous DNA by following a crRNA-guided mechanism. Further additional processing requires trans-activating CRISPR RNA (tracrRNA) [35]. The tracrRNA–crRNA–Cas ribonuclease complex can recognize and cleave any exogenous DNA molecule that shares homology with the crRNA (Figure 1c). Another key element necessary for Cas cleavage activity is the so-called protospacer-adjacent motif (PAM) sequence, a conserved sequence of 2–6 base pairs flanking the target DNA sequence. Consequently, a sequence with perfect complementarity to the crRNA guide but lacking a PAM will be ignored by the Cas nuclease [36].

Soon after the molecular mechanism was uncovered, several research groups were able to edit genes in different organisms using the CRISPR/Cas system with great success [37,38,39,40,41]. A major step forward was the creation of a chimeric molecule called single guide RNA (sgRNA), which is a fusion of crRNA and tracrRNA and contains a short sequence complementary to the endogenous sequence of the gene to be edited. By varying the sgRNA sequence, it becomes possible to target different regions of DNA. Different CRISPR/Cas systems have been thoroughly investigated [42]. Two major classes, namely class 1 and class 2, comprise multi-subunit effector complexes or single-protein effector complexes, respectively, which are further divided into types (I, III, IV and II, V, VI, respectively) and subtypes [42,43]. Of the various Cas proteins identified, the most frequently used for gene editing are type II-A Cas9 from *Streptococcus pyogenes* [42] and type V-A Cas12a (Cpf1) from *Acidaminococcus* sp. And *Lachnospiraceae bacterium* [44].

#### 2.3.1. Cas9

The most common method of gene editing utilizes the Cas9 protein of *Streptococcus pyogenes*. The Cas9 protein is a dual RNA-guided DNA endonuclease able to cleave the target DNA in both strands through recognizing the hairpin loop architecture of the crRNA–tracrRNA complex [45]. Cas9 guided by multiple sgRNAs can bind to endogenous genes with high specificity. In 2013, the first CRISPR/Cas9 edited plants of *Arabidopsis thaliana*, *Nicotiana benthamiana*, and *Oryza sativa* were obtained [46,47]. Different versions of Cas9 with high efficiency for gene editing have been identified and utilized in plants since then [48].

When compared to other gene-editing approaches, the CRISPR/Cas9 system exhibits high rates of editing efficiency, approximately 75% to 85% [49,50,51,52], surpassing those of ZFNs and TALENs [53]. However, the CRISPR/Cas9 system for gene editing has several drawbacks, including reduced editing efficiency due to PAM sequence mismatches, genetic mosaicism, and editing of non-targeted regions of the genome (i.e., off-target gene editing). To address these issues, a mutant version of Cas9 called Cas9 nickase (nCas9) has been engineered to generate single-strand DNA breaks and reduce off-target effects [54].

#### 2.3.2. Cas12

Generally, Cas12a and their orthologs are smaller than Cas9 as they possess the nuclease domain but lack the helicase domain (Figure 1d). Certain Cas12a proteins serve as single effectors, which induce DSBs on the target DNA and are guided by smaller crRNAs (~42 nt) in contrast to those of Cas9 crRNAs (~100 nt) [44,55]. Cas12a has the ability to process pre-crRNA into mature crRNA independently of the tracrRNA. Additionally, there are other unique features of Cas12a proteins that distinguish them from Cas9. These include their preference for thymidine-rich PAM, whereas Cas9 prefers guanidine-rich sequences. Furthermore, after Cas12a cleaves the target DNA, it produces a 5′ overhang, increasing the efficiency of precise gene editing in AT-rich target regions, which Cas9 has difficulty accessing [44]. Beyond that, while Cas9 and Cas12a exhibit similar mismatches in their target DNA when assessed in vitro [56], it has been observed that Cas12a displays a lower off-target effect than Cas9 when conducting gene editing [57,58]. Initial successful outcomes following gene editing in plants using Cpf1, which is the Cas12a ortholog from *Francisella novicida*, were achieved in 2016 for *Oryza sativa* and *Nicotiana benthamiana* [59].

#### 2.3.3. Other Cas Proteins

Cas13a is an RNA-guided RNA endonuclease from the bacterium *Leptotrichia shahii* that does not cleave DNA, but only single-stranded RNA [60]. Its use has been proposed to engineer resistance against plant RNA viruses and regulate gene expression post-transcriptionally [61]. Additionally, a few recent studies demonstrated that the hypercompact CRISPR/Cas gene-editing system of Cas12j encoded in some phage genomes can be used for efficient gene editing in plants with expanded target recognition capabilities compared to other Cas proteins [62,63,64].

### 2.4. Other Gene-Editing Tools

Several other gene-editing tools have been developed utilizing Cas9’s ability to bind to specific DNA sequences via their sgRNAs. These methods all employ non-functional variants of the Cas9 nuclease, also referred to as dead Cas9 (dCas9). dCas9 binds to the sgRNA target site on the DNA but does not cause DSBs, instead interfering with downstream transcription [65]. The fusion of specific deaminases with dCas9 enables the engineering of base editors, which direct the introduction of point mutations in specific sequences of a target DNA without initiating DSBs [66]. There are two classes of DNA base editors, namely cytosine base editors (CBEs) and adenine base editors (ABEs). ABEs convert the A=T base pair into G≡C, whereas CBEs convert G≡C into the A=T base pair [48,67]. Base editing has been successfully applied to a range of agronomic traits in crop plants, including wheat, rice, and tomatoes [68,69]. In prime editing, the dCas9 is fused with a reverse transcriptase (RT) enzyme, allowing for the introduction of base changes at the targeted site via the RT activity and using a prime editing guide RNA (pegRNA). This pegRNA not only specifies the target site but also encodes the desired edit [70]. Efficient cases of prime editing were previously reported for monocots, including rice, wheat, and maize, followed by mosses and dicots [71].

The different gene-editing methods described in this section allow for the precise modification of specific regions of the plant genome. CRISPR/Cas systems are the most widely used gene-editing tool due to their simple design, low cost, high efficiency, good repeatability, and short cycle time. Nevertheless, it is important to consider the advantages and disadvantages of ZFNs, TALENs, and CRISPR/Cas systems for gene editing to improve crop performance (Table 1).

## 3. Methods for Obtaining Gene-Edited Plants

### 3.1. Methods for CRISPR/Cas Delivery

In order to induce DSBs and activate the endogenous DNA repair mechanism, which ultimately leads to gene editing, the Cas protein and sgRNA(s) components of the CRISPR/Cas system must reach the nucleus of the plant cell. This can be achieved through conventional delivery of genetically encoded CRISPR/Cas components or through preassembled gRNA/Cas ribonucleoprotein (RNP) complexes. The RNP complexes target and cleave the target sites in the DNA immediately after delivery and are rapidly degraded in those cells. Several recent systematic reviews have described the different methods used to deliver the CRISPR/Cas system to plants [72,73,74,75], and we will not provide an exhaustive description of them here. Below, you will find a brief overview of these methods, with an emphasis on recent developments for CRISPR/Cas delivery in tomatoes.

#### 3.1.1. Particle Bombardment

The biolistic method delivers molecules into plant nuclei using accelerated nano-sized particles coated with nucleic acids or RNP complexes (Figure 2a) [76]. However, there are several disadvantages to this approach, including the random integration of the cargo DNA at multiple sites in the genome and the high cost of the equipment and reagents. Regenerating whole plantlets from transformed explants is a time-consuming process that is also dependent on the species and the genotype. RNP-mediated gene editing using particle bombardment has been successfully achieved in several species, but it has rarely been reported in tomatoes [77].

#### 3.1.2. Polyethylene Glycol (PEG)-Mediated Transfection

In the presence of divalent cations, such as Ca^2+^ or Mg^2+^, and at high pH, PEG facilitates the incorporation of exogenous DNA or RNP complexes into plant protoplasts, but with low transformation efficiency (Figure 2b). This approach relies on the establishment of successful protoplast isolation and regeneration procedures. PEG-mediated transfection has been widely used to deliver vector DNA or RNPs into protoplasts of many plant species, including various tomato cultivars [78,79,80,81] and the wild tomato *S. peruvianum* [82].

#### 3.1.3. Biological Methods

Biological methods for delivering DNA into plant nuclei rely on the natural ability of plant-pathogenic soil bacteria of the genus *Rhizobium* to integrate some of the genes present in their T-DNA into the genomic DNA of plant cells [83,84]. *A. tumefaciens*, also known as *Rhizobium radiobacter*, is the most commonly used species for delivering vectors encoding the CRISPR/Cas components. Conversely, *A. rhizogenes*, also known as *R. rhizogenes*, has received considerable attention in recent years during studies of root-specific processes [85]. A large number of laboratory strains of *A. tumefaciens* are available to transform a wide variety of plant species with varying success [86]. Recently, the *A. tumefaciens* and *A. rhizogenes* genomes were modified to improve their transformation efficiency and broaden their host range of plant species [87,88].

*Agrobacterium* spp. cell cultures containing the desired vectors in the exponential growth phase are used to transform plant tissues (Figure 2c). For plant species such as *A. thaliana*, the floral dip method [89] is preferred, while in most other cases, *A. tumefaciens* must be co-cultured with tissue explants, such as cotyledons, leaves, calli, or embryos. In such cases, a suitable protocol for regenerating whole plants from the few transformed cells with the T-DNA from *A. tumefaciens* is required. In the case of tomatoes, cotyledons can be easily transformed through co-culturing with various *Agrobacterium* spp. strains. Shoot induction can then be produced shortly afterwards with appropriate hormone combinations [90].

Transient expression of CRISPR/Cas components can be induced in plant tissues by delivering *A. tumefaciens* into plant cells through direct injection or vacuum infiltration (Figure 2d,e1) [91]. This method is appropriate for evaluating the gene-editing efficiency of different vectors in tomato leaves before conducting stable transformations [92,93]. Zhang and coauthors (2020) tested 195 sgRNAs for their ability to cause mutations in tomato leaves, and they found that 61.5% of them were suitable for gene editing [92].

Virus-induced gene editing (VIGE) relies on the use of viral-derived vector systems for the delivery of CRISPR/Cas components into plant cells [94]. The VIGE approach does not require genome integration of the CRISPR/Cas backbone, making it less time-consuming and more broadly applicable. However, the primary limitation of this approach is that most viruses are unable to infect meristematic or germline cells. As a result, regenerated edited plants are normally derived from other somatic cells [94]. Nonetheless, various approaches based on vectors derived from geminivirus [95] and potato virus X [96] have been successfully employed to produce stable edited tomato plants using direct injection (Figure 2e2). The results suggest that the VIGE approach is a reliable and adaptable technology that can be used for precise breeding of tomato traits.

#### 3.1.4. Other Methods of CRISPR/Cas Delivery

Liposomes are lipid-based nanoparticles that have been shown to offer a useful tool for the delivery of genes and proteins (Figure 2f) into mammalian cells [97] and that have been used to deliver donor DNA and support subsequent successful gene editing in citrus protoplasts [98]. In addition, RNPs have been delivered in tobacco protoplasts through liposome nanoparticles, achieving editing efficiencies as high as 6.0% [99]. Moreover, recently, a similar protocol has been devised for the successful delivery of RNPs in maize protoplasts [100].

Electroporation, meanwhile, is a technique that increases the permeability of cell membranes in plant cells without cell walls (protoplasts) by exposing them to short and intense electric pulses (Figure 2g). This allows for the incorporation of exogenous nucleic acids or RNPs [101]. Recently, researchers have demonstrated the delivery of RNPs into cabbage and soybean protoplasts via electroporation, achieving editing efficiencies as high as 3.4% and 8.1%, respectively [102,103]. Electroporation has also been used in tomatoes for the delivery and transient expression of vectors encoding ZFNs [104], but no other reports with CRISPR/Cas vectors of RNPs can be found in this species so far.

### 3.2. Marker Genes Used with the CRISPR/Cas System

When using conventional CRISPR/Cas vectors, most gene-edited plants include the stable integration of the CRISPR/Cas system into the plant genome. Furthermore, the CRISPR/Cas expression module typically includes genes encoding selection markers for antibiotic or herbicide resistance. However, the presence of the transgene does not necessarily indicate that gene editing has taken place. Characterizing the mutations that have occurred in the target DNA often requires laborious procedures, including high-resolution melting PCR or Sanger sequencing. Additionally, to conform with biosafety regulations concerning genetically modified organisms (GMOs), the CRISPR/Cas expression module must be segregated in the progeny of gene-edited plants through self-pollination or crossing. Alternatively, the delivery of preassembled CRISPR/Cas components is considered a transgene-free method of gene editing. In this approach, the RNP complexes have high specificity for their target DNAs and are rapidly degraded within the cell after dissociating from the genomic DNA. The fusion of Cas proteins with fluorescent proteins, such as enhanced GFP (EGFP), enables the direct visualization of the RNP complexes in plant nuclei, which is highly convenient during the optimization of the delivery process.

For both approaches, it is beneficial to select a CRISPR/Cas-induced mutation that causes a visible phenotype. A frequently used endogenous target gene is the *PHYTOENE DESATURASE* (*PDS*) gene *Solyc03g123760*, which is involved in carotenoid biosynthesis. Mutations in this gene impair chlorophyll accumulation and produce an albino phenotype in many species, including tomatoes [105]. However, mutations in *PDS* affect the survival of the edited explants, which hinders the effective plant regeneration required for successive characterization. Recently, Rinne et al. (2021) proposed using a different endogenous marker gene to evaluate CRISPR/Cas gene-editing activity [106]. The *MULTIPLE ANTIBIOTIC RESISTANCE 1* (*MAR1*) gene *Solyc01g100610* encodes a transporter located in the mitochondria and chloroplasts, which plays a role in maintaining iron homeostasis and facilitates the transport of aminoglycoside antibiotics into these organelles. Mutations induced by the CRISPR/Cas system in *MAR1* confer kanamycin resistance in tomatoes [106].

## 4. Gene Editing in Tomato Breeding

### 4.1. ZFNs

Only a limited number of studies are available on the application of ZFNs to edit tomato plants [107,108]. Shukla et al. (2016) conducted an experiment where they utilized ZFNs to target *Solyc07g062650*, which encodes a mitochondrial malate dehydrogenase (mMDH), in both the M82 and Moneymaker tomato cultivars [107]. PEG-mediated transfection was used to deliver four distinct ZFN constructs into protoplasts, and the regenerated plantlets were obtained via indirect organogenesis. The editing efficiency of *mMDH* was found to be low, ranging from 0.7% to 5.5%. The majority of mutants were found to have small deletions varying from 1 to 22 bp, while one mutant showed an insertion of 2 bp. Regenerated mutants displayed different phenotypes: heterozygous plants for any mutation that led to disruption of the *mMDH* open reading frame showed a decrease in growth and fruit yield, while homozygous plants for specific mutations exhibited an increase in fruit yield [107]. In another study, ZFNs were used to target the *LEAFY-COTYLEDON1-LIKE4* (*L1L4*) gene *Solyc05g005350*, which encodes for the β-subunit of a nuclear factor Y, in the Heinz 1706 cultivar [108]. Electroporation of germinated seeds facilitated the transient expression of constructs encoding a pair of ZFNs designed to target exons 1 and 2 of the *L1L4* gene. Based on the phenotypes observed in the edited plants from the T_0_ generation, over 65% efficiency was reported. Several mutations in the coding sequence of *L1L4* have led to variations in plant traits, including seedling vigor, plant height, number of florets, and flowering and ripening times, thereby demonstrating that the affected *L1L4* gene is a useful target for crop improvement [108].

### 4.2. TALENs

Three reports have been published to date on the use of TALENs for tomato gene editing. Lor et al. (2014) designed two pair of TALENs, under the control of an estrogen-inducible promoter, to target the *PROCERA* (*PRO*) gene in tomatoes, *Solyc11g011260* [109]. This gene encodes a negative regulator of gibberellin signaling. Upon TALEN expression, 7 out of 40 regenerated plantlets carried deletions ranging from 1 to 88 bp, along with a 39-bp insertion and a 4-bp deletion, resulting in frameshifts that would lead to the production of truncated PRO proteins. Homozygous *pro* plants displayed phenotypes indicating an increased gibberellin response [109]. In another report, geminivirus replicons were used to express TALENs designed to target the *ANTHOCYANIN 1* (*ANT1*) gene *Solyc10g086260*, encoding an MYB transcription factor whose overexpression results in purple plant tissue due to anthocyanin accumulation. The gene-editing efficiency of this system was about 14% [95]. In a recent report, a known mutation in the pepper gene that encodes *EUKARYOTIC TRANSLATION INITIATION FACTOR 4E* (*eIF4E*), *Solyc03g005870*, which is reported to be associated with potyvirus resistance, was generated through a gene knock-in strategy with a pair of TALENs in tomato [110]. The TALEN vector and the donor DNA template were introduced to cherry tomato leaves in a biolistic approach. One out of thirty-two regenerating shoots incorporated the donor template through NHEJ, and the edited plants with this allele showed the broadest potyvirus resistance spectrum achieved through genetic engineering in tomatoes so far [110].

### 4.3. CRISPR/Cas9

The CRISPR/Cas9 approach was first applied to tomato gene editing in 2014 [111,112]. As a proof of principle, Brooks et al. (2014) designed two sgRNAs to target the tomato homolog of the *ARGONAUTE7* (*AGO7*) gene *Solyc01g010970* [111], since loss-of-function mutations of *AGO7* are known to produce filiform leaves [113]. The plasmid vector was delivered via *Agrobacterium tumefaciens* transformation of M82 cotyledons, and 14 out of 29 T_0_ plants displayed the wiry leaf phenotype characteristic of known *ago7* alleles [111]. Ron et al. (2014), meanwhile, used *A. rhizogenes*-mediated transformation of cotyledon explants as a delivery method for the CRISPR/Cas9 vector, which was designed to target both a reporter gene (*mGFP5*) and the *Solyc02g092370* gene encoding a *SHORT-ROOT* (*SHR*) protein in tomato hairy roots [112]. Consistent with the notion of a conserved role of SHR in both tomatoes and *A. thaliana*, transgenic hairy roots with no GFP expression and a reduced root meristem were identified. These roots contained small insertions and deletions in the coding region of the *SHR* gene [112].

In the past decade, the use of CRISPR/Cas9 for gene editing in tomatoes has been implemented in dozens of tomato cultivars [114,115], including wild species such as *S. lycopersicum* var. *cerasiforme* [110], *S. pimpinellifolium* [116,117,118,119,120,121], and *S. peruvianum* [82]. When searching the scientific literature available as of 31 December 2023, we identified 356 primary references that used CRISPR/Cas technology for gene editing in tomato and related species (Figure 3a–c and Appendix A). From this list, we selected a subset of references to be further discussed (Appendix A). A Sankey diagram constructed from the evidence presented in these selected references illustrates the relationship between tomato cultivars, their wild relatives, Cas enzyme types, *A. tumefaciens* strains, and the key genetic traits influenced by the genetic modifications induced by CRISPR/Cas9 (Figure 3d).

Several studies have documented the domestication process from the most likely progenitor, *S. pimpinellifolium*, to modern tomato varieties [122]. Loss-of-function mutations in six loci that are important for key domestication traits have been identified from these and other studies: plant architecture (*SELF PRUNING* [*SP*] *Solyc06g074350*), fruit shape (*OVATE* [*O*] *Solyc02g085500*), fruit size (*FASCIATED* [*FAS*] *Solyc11g071380*, and *FRUIT WEIGHT 2.2* [*FW2.2*] *Solyc02g090730*), fruit number (*MULTIFLORA* [*MULT*] *Solyc02g077390*), and nutritional quality (*LYCOPENE BETA CYCLASE* [*CycB*] *Solyc04g040190*). To reconstruct the domestication of tomatoes from *S. pimpinellifolium*, Zsögon et al. (2018) [121] designed a CRISPR/Cas9 vector containing six sgRNAs that targeted specific sequences in the coding regions of these six genes. Four of the six targeted loci were successfully edited in all 50 T_1_ lines tested, resulting in indel mutations in *SP*, *O*, *FW2.2*, and *CycB* [121]. In a subsequent round of gene editing, the researchers incorporated *CLAVATA 3* (*CLV3*) *Solyc11g071380*, and developed a new CRISPR/Cas9 vector with eight sgRNAs targeting *CLV3*, *FW2.2*, *MULT*, and *CycB* genes. This resulted in 28 T_1_ lines with loss-of-function mutations in all four targeted loci [121]. The engineered lines exhibit a threefold increase in fruit size and a tenfold increase in fruit number when compared with the wild parent, *S. pimpinellifolium*. A significant finding was that the accumulation of fruit lycopene in these lines was enhanced fivefold in comparison to the cultivated tomato *S. lycopersicum* [121]. Similarly, Li et al. (2018) developed a multiplex CRISPR/Cas9 strategy to edit five genes in *S. pimpinellifolium* related to day-length sensitivity (*SELF PRUNING 5G* [*SP5G*] *Solyc05g053850*), shoot architecture (*SP*), flower and fruit production (*CLV3* and *WUSCHEL* [*WUS*] *Solyc02g083950*), and vitamin content (*GGP1 Solyc02g091510*) [119]. Furthermore, Lemmon et al. (2018) used CRISPR/Cas9 to mutate orthologs of tomato domestication genes that control plant architecture (*SP*), flower production (*SP5G*), fruit size (*CLAVATA1*), and fruit abscission (*JOINTLESS-2*) in the orphan crop *Physalis pruinosa* [123]. This wild *Solanaceae* is distantly related to tomato and is grown in Central and South America for its subtly sweet berries. Overall, the results suggested that CRISPR/Cas9-mediated gene editing could be used to accelerate domestication in wild species and create crops that are better suited to changing climate scenarios [124,125].

#### 4.3.1. CRISPR/Cas9-Edited Genes Related to Fruit Characteristics

Tomato plants are grown for their fleshy berry fruits, which come in varying sizes, shapes, and colors [126,127]. Tomatoes contain several health-promoting components, including vitamins, carotenoids, and phenolic compounds [128]. Tomato breeders have long utilized the limited genetic variation present in this species to enhance fruit quality traits. Some of the genes responsible for variations in fruit weight, shape, and biochemical composition have been identified and those might be used for precise gene editing using the CRISPR/Cas9 system (Appendix A), as will be exemplified below.

Tomato fruit ripening is a complex physiological and biochemical process that involves the degradation of chlorophyll, the accumulation of pigments (mainly carotenoids), the softening of the pulp, and the alteration of its organoleptic qualities. This process is controlled by ethylene, and requires three transcription factors: the MADS-box RIPENING INHIBITOR (RIN), the SBP-box COLORLESS NON-RIPENING (CNR), and the NAC transcription factor NON-RIPENING (NOR) [129]. One of the initial reports on the application of CRISPR/Cas9 technology in tomatoes targeted the *RIN* gene (*Solyc05g012020*) [130,131]. Fruits from *RIN*-knockout plants exhibited partial ripening and lower levels of lycopene compared to the wild type [131]. However, unexpectedly, these fruits showed excessive cell wall degradation. Furthermore, CRISPR/Cas9-induced mutations in *CNR* (*Solyc02g077920*) and *NOR* (*Solyc10g006880*) did not abolish ripening [132,133], indicating that the ripening transcriptional regulatory network is highly robust. These results are in stark contrast with those found in “classical” *rin*, *Cnr*, and *nor* mutants, which have obvious ripening-inhibited phenotypes. The difference in findings may be explained by the recent proposal that some of these mutants have gain-of-function mutations [134].

The color change that occurs during fruit ripening is due to the accumulation of β-carotene (orange) and lycopene (red), as well as a decrease in β-xanthophyll (yellow) and chlorophyll (green) levels [135]. Furthermore, the expression of *SNAC4* (*Solyc07g063420*) and *SNAC9* (*Solyc04g005610*) genes is induced by ethylene, and alterations in the pigmentation of mature fruits are observed when the expression levels of these genes are reduced by VIGS, indicating a positive role of both genes in this process [136]. In previous research, the *SNAC9* gene was knocked out thoroughly using CRISPR/Cas9, resulting in an absence of expression of the protein. This led to a significant reduction in lycopene and total carotenoid contents in the mutants, possibly due to the direct regulation of key genes involved in carotenoid biosynthesis, such as *PHYTOENE SYNTHASE 1* (*PSY1*) *Solyc03g031860* [137]. A strategy using NGTs has recently been implemented to rapidly generate tomato lines with fruits of varying colors [135,138,139]. By genetically editing three genes related to fruit color, *PSY1*, *MYB12* (*Solyc01g079620*), and *STAY-GREEN 1* (*Solyc08g080090*), using a multiple CRISPR/Cas9 approach, it was possible to obtain transgene-free plants with fruits of different colors in less than a year. This strategy does not affect other important agronomic traits, such as yield and fruit quality [138].

Initial attempts to extend the shelf life of tomato fruits using CRISPR/Cas9 were focused on genes that encode enzymes responsible for degrading cell-wall pectins, such as pectate lyase (PL), polygalacturonase, and β-galactanase. However, only mutations in the *PL* gene *Solyc03g111690*, resulted in firmer fruits [140]. Simultaneous knockout of two genes, one encoding a fruit ripening-associated α-expansin (*Solyc06g051800*) and the other an endoglucanase (*Solyc09g010210*), using CRISPR/Cas9 improved fruit firmness [141]. Conversely, simultaneous overexpression of these genes promoted early fruit softening. These results support the conclusion that these two genes have a synergistic effect on fruit softening [141]. Furthermore, the bHLH transcription factor BRI1-EMS-SUPPRESSOR1 (BES1) encoded by *Solyc04g079980* has been identified as an upstream regulator of fruit softening through the activation of cell-wall degradation during ripening [142]. CRISPR/Cas9-generated loss-of-function *BES1* mutants resulted in firmer fruits and a longer shelf life during postharvest storage, without any negative impact on visual or nutritional quality.

#### 4.3.2. CRISPR/Cas9-Edited Genes Related to Plant Architecture Traits

Modifying the plant architecture can significantly impact crop yield. This was demonstrated through mutations to the biosynthesis or signaling pathways of gibberellins, which increased crop yield by decreasing plant height in wheat and rice, an outcome that is of value during the “Green Revolution” [143]. Furthermore, in rice, the *IDEAL PLANT ARCHITECTURE 1* (*IPA1*) gene has been determined to be a novel master regulator of the plant architecture, which can be used as a target during molecular design to improve grain yield. *IPA1* encodes an SBP-box-like (SPL) transcription factor downstream of strigolactone signaling [144]. One of the *IPA1* orthologs of tomatoes, *SPL13 Solyc05g015840*, has been shown to act downstream of strigolactones to suppress lateral bud growth by inhibiting cytokinin biosynthesis. The knockout lines of *SPL13* generated with CRISPR/Cas9 displayed an increased shoot branching phenotype [133]. In other work, Kwon et al. (2020) developed a gene-editing staking approach to transform vine-like tomato plants into compact, early-yielding plants suitable for indoor agriculture [145,146]. Elsewhere, mutations in the classical flowering repressor gene *SP*, both natural and CRISPR/Cas9-induced, conferred a determinate growth habit in tomatoes [121]. Furthermore, mutating its paralog *SP5G* with CRISPR/Cas9 in the *sp* mutant background resulted in a shorter time to flowering and a more compact, determinate growth habit. This led to a quick burst of flower production and an early yield [147]. Meanwhile, cloning the *short internodes* dwarf mutant in the M82 cultivar was found to affect the well-known *ERECTA* (*ER*) gene [145]. In *A. thaliana*, this gene is known to regulate internode length [148]. The specific *ER* gene *Solyc08g061560* was targeted using CRISPR/Cas9 in the previously generated *sp sp5g* double mutant [147]. The resulting plants exhibited a triple-determinate form, and when grown in a self-contained, climate-controlled LED hydroponic vertical farm system, they produced higher yields due to their bushy shoot architecture, rapid cycling, and highly compact fruit clusters [145].

Naturally occurring mutations in the MADS-box transcription factor genes *JOINTLESS* (*J*) and *JOINTLESS2* (*J2*) in tomatoes have the potential to improve harvestability by modifying flower development, as they suppress the abscission zone in fruit peduncles [149]. The jointless pedicel trait has been successfully introgressed into small-fruited processed tomatoes and fresh-market tomatoes. However, the *j2* loss-of-function mutation can lead to undesirable branching of inflorescences in genetic backgrounds that also carry a cryptic variant for the close homolog *ENHANCER OF J2* (*EJ2*) *Solyc03g114840*, which was selected during domestication. This combination of *j2 ej2* loss-of-function alleles results in excessive flower production and low fertility due to a poor fruit set [150]. Deletions induced by CRISPR/Cas9 in *J2 Solyc12g038510*, result in the jointless phenotype. When combined with CRISPR/Cas9-induced null mutations in *EJ2*, this leads to strongly branched inflorescence in tomato breeding lines carrying the *suppressor of branching 3* (*sb3*) quantitative trait loci (QTL). These results suggest that genotypes carrying *sb3* could be utilized to maintain a normal inflorescence architecture when generating jointless phenotypes via gene editing [150].

During the process of tomato domestication, natural genetic variants were selected based on their alterations in the expression of key genes involved in target agronomic traits, such as yield or plant architecture. In many cases, the molecular characterization of these genetic variants has shown that they do not affect the coding region of the gene, but rather its *cis*-regulatory regions, either in gene promoters or in regulatory introns. Structural variations, such as large indels, duplications, and chromosomal rearrangements, play a crucial role in plant evolution and agriculture. They impact traits such as shoot architecture, flowering time, fruit size, and stress resistance [151]. Gene editing may be used to study the effects of variations in regulatory regions by recreating specific mutations or mimicking the expression effects of natural *cis*-regulators, which could be achieved by using the CRISPR/Cas approach on defined tomato genotypes. Accordingly, Rodríguez-Leal et al. (2017) developed a genetic screen that uses heterozygous loss-of-function mutant backgrounds to efficiently assess the phenotypic effects of multiple CRISPR/Cas9-induced promoter variants for known genes that regulate three key productivity traits in tomato: fruit size (*CLV3*, *WUS*), inflorescence branching (*MULT*), and plant architecture (*SP*) [117]. The promoter alleles for *CLV3* exhibited a full range of quantitative variation. The edited plants displayed phenotypes ranging from moderate to strong, capturing the full spectrum of allelic diversity and locule number variation previously identified [117]. This approach has been demonstrated to generate multiple regulatory mutations for the systematic assessment of the association between *cis*-regulatory regions and phenotypic variation. Furthermore, there is also potential for engineering gain-of-function alleles and thereby establishing a basis for dissecting the complex relationships between gene-regulatory alterations, which may facilitate quantitative trait control.

#### 4.3.3. CRISPR/Cas9-Edited Genes Related to Adaptive Stress Responses

Systematic reviews have compiled extant studies on gene editing with CRISPR/Cas9 in agricultural crops in the scientific literature [152,153,154]. Various biotic stresses, such as diseases and pests, as well as abiotic stresses such as cold, heat, drought, and salinity, affect tomato crop production, and Appendix A shows that CRISPR/Cas9 gene editing has been used to address many of these traits. Those applications will be summarized in this section.

The *MILDEW RESISTANCE LOCUS O* (*MLO*) encodes an integral transmembrane protein that is highly conserved in monocots and dicots. It is a key factor in susceptibility to powdery mildew caused by the pathogenic fungus *Oidium neolycopersici*. In cultivated tomato and the wild species *S. peruvianum*, CRISPR/Cas9-induced knock-out lines of *MLO1* orthologs *Solyc04g049090* and *pSolyc04g049090* conferred resistance to *O. neolycopersici* without generating any other undesired phenotypic effects [82,155,156]. Furthermore, the *DOWNY MILDEW RESISTANT 6* (*DMR6*) gene in *A. thaliana* encodes a putative 2-oxoglutarate Fe(II)-dependent dioxygenase that has been identified as a factor in susceptibility to bacterial and oomycete pathogens. The tomato genome contains two orthologs of *DMR6*, *DMR6-1* (*Solyc03g080190*) and *DMR6-2* (*Solyc06g073080*). Inactivation of *DMR6-1* using CRISPR/Cas9 resulted in enhanced disease resistance against different tomato pathogens, such as bacteria, oomycetes, and fungi, without an obvious growth penalty [157]. Notably, the enhanced pathogen resistance observed in the *dmr6-1* tomato mutants was correlated with increased levels of salicylic acid. The biochemical characterization of DMR6 enzymes suggests that they play a role in converting salicylic acid to its inactive form [157]. In other work, the CRISPR/Cas9 system has been used to mutate the susceptibility gene *POWDERY MILDEW RESISTANCE 4* (*PMR4*) *Solyc07g053980*, with the product functioning as a callose synthase conferring resistance to *O. neolycopersici* and *Phytophtora infestans* in tomatoes [158,159]. These results demonstrate that the CRISPR/Cas9 system is suitable for facilitating broad resistance to bacterial and fungal pathogens by precisely targeting susceptibility genes and negative regulators involved in the plant defense mechanism.

Tomato yellow leaf curl virus (TYLCV) is a highly destructive viral pathogen that affects tomato crops globally. TYLCV is transmitted by the whitefly *Bemisia tabaci* when it feeds on the phloem sap of plants. Previous strategies to confer resistance to TYLCV in tomato plants have focused on the viral coat protein and DNA replicase protein, CP and RepA, respectively. Efficient viral interference was achieved by targeting the TYLCV genome with CRISPR/Cas9 for these sequences, resulting in reduced TYLCV accumulation in transgenic tomato plants [160]. Several QTL related to TYLCV resistance in tomatoes, including *Ty-1* to *Ty-6*, have been identified [161]. *ty-5* confers broad-spectrum resistance and encodes a missense allele of the tomato homolog of the PELOTA messenger RNA surveillance factor, which is involved in ribosome recycling during protein translation. Gene editing of the *PELOTA* gene *Solyc04g009810*, using a CRISPR/Cas9 approach resulted in resistance to TYLCV as a result of restricting the proliferation of viral DNA, likely by slowing or inhibiting ribosome recycling and decreasing viral protein synthesis in infected cells [156]. Targeting susceptibility factors encoded by the host plant genome, rather than the viral genome, offers a promising approach to achieving resistance to pathogens without the need for stable inheritance of CRISPR/Cas9 components.

In the context of global warming, drought stress is becoming a critical challenge in tomato production. As previously discussed [162], the development of new CRISPR/Cas9-based approaches to improve drought tolerance in tomatoes is essential to reduce yield loss. Several key factors have been considered (Appendix A). Among those, in a scenario where the tomato *LATERAL ORGAN BOUNDARIES DOMAIN 40* (*LBD40*) *Solyc02g085910* was highly expressed in the roots and fruits and its expression was significantly induced by PEG and salt treatment, *LBD40* knockout mutants generated by CRISPR/Cas9 gene editing improved the water-holding ability of tomatoes under drought conditions, suggesting that *LBD40* was a negative regulator of drought tolerance in this species [163]. Furthermore, two studies have highlighted the role of *AUXIN RESPONSE FACTOR 4* (*ARF4*) *Solyc11g069190* in mediating the tolerance to salinity and osmotic stress in tomatoes [164,165]. The expression of the *ARF4* gene was downregulated in tomato seedlings under ABA and water deficit conditions. Downregulating the expression of *ARF4* with CRISPR/Cas9 was observed to enhance salt and osmotic stress tolerance recovery. This resulted in an obvious leaf curling phenotype, which reduced transpiration, and a significant increase in root length and density compared to wild type plants under stress conditions [164,165]. More recently, the expression of *SP2G* (*Solyc02g079290*) and *SP3C* (*Solyc03g026050*) was found to be significantly increased in the leaves of the drought-resistant wild species *S. pennellii* and in domesticated tomatoes after irrigation was stopped [166]. Furthermore, three independent CRISPR/Cas9-mediated *SP3C* homozygous mutants exhibited increased root length and reduced lateral root branching compared to the wild type. These traits are associated with greater drought tolerance in tomatoes and could be fine-tuned for agronomic gain [166]. Further research into the mechanisms of action of key transcription factors that regulate the abiotic stress response in tomatoes may expand the resources and tools at our disposal for developing multi-stress tolerant tomato varieties.

### 4.4. CRISPR/Cas12a

Two different Cas12a variants were used to stably transform tomato plants so they targeted *Solyc01g079620*, which encodes the MYB12 transcription factor required for flavonoid biosynthesis. A mutation in this gene leads to the production of pink-colored fruits [167]. The gene-editing efficiency of this system ranged from 7.7% to 48.8%, with a tendency to induce a broader range of deletions but no insertions, in contrast to Cas9 [168]. Furthermore, as demonstrated by Vu et al. (2020), the CRISPR/Cas12a system has the advantage of an increased efficiency in genome editing via the homology-directed repair (HDR) pathway when compared to Cas9. The authors reported an HDR efficiency of 4.5% of the *ANT1* visible marker when using a geminiviral replicon system [169]. In another study, to confirm the applicability of the Cas12a-mediated HDR approach for tomato genome editing of a potential agronomic trait, a known mutation in the *HIGH-AFFINITY K^+^ TRANSPORTER 1;2* (*HKT1;2*) gene *Solyc07g014680*, which determines salinity tolerance [170], was engineered. However, the efficiency of HDR was low, at only 0.7%. The edited plants were salt-tolerant in both homozygous and heterozygous states [169]. In other work, a higher HDR efficiency of 4.3% was reported when using a new Cas12a variant that is capable of performing the cleavage of the target DNA at low temperatures [171]. Together, these results confirm that Cas12a-mediated HDR induces efficient gene targeting, which may be used to obtain allele-specific traits and marker-free tomato plants. More recently, Cas12 has been fused to a CBE with the aim of conferring resistance to the herbicide chlorsulfuron via precise editing of the *ACETOLACTATE SYNTHASE* (*ALS*) gene *Solyc03g044330* [172]. More than 20 chlorsulfuron-resistant lines were obtained in this way and the mutations were confirmed to be highly specific. In addition, the authors edited the *Solyc08g061560* gene, which encodes the ER kinase receptor. Mutations in this gene in *A. thaliana* resulted in a compact inflorescence [148]. The phenotype of the homozygous edited tomato plants in this gene included a compact architecture, short petioles, and densely clustered inflorescence [172].

## 5. Conclusions

Over the last 15 years, significant progress has been made in crop gene editing using the NGTs of ZFNs, TALENs, and CRISPR/Cas. Among these, the CRISPR/Cas-based editing system has emerged as the preferred option due to its many benefits. The high efficiency of gene editing induced by the Cas proteins, together with the flexibility in the design of gene-specific RNA guides, allows the chosen experimental design to be refined while considering the genotype of the tomato variety, thus supporting the ultimate objective pursued with the genetic modification. A wealth of evidence supports the utility of the CRISPR/Cas system for the genetic editing of multiple genes in different varieties of cultivated tomatoes and related wild species. In this review, we conducted a systematic search of the references and identified 356 scientific articles that reported primary results on the modification of a gene using NGTs in cultivated tomatoes or related wild species. From these, we selected 47 genes that affect key genetic traits that we determine to warrant further discussion. Some of these genes were found to impact the plant architecture, resulting in increased shoot and flower branching, a more compact growth habit, and a shorter flowering time. Engineering such variations can thus facilitate the development of new tomato varieties that are better suited to indoor farming and have higher crop yields than those at present. Furthermore, we have discussed tomato plants that have been gene edited to increase their tolerance to various pathogens and viruses, including *Oidium neolycopersici*, *Phytophtora infestans*, and TYLCV. Additionally, gene-edited tomatoes are being developed to exhibit greater tolerance to drought or salt/osmotic stress.

The widespread application of CRISPR/Cas methodologies for precise genetic modification will facilitate the development of tomato plants that are more tolerant to the multiple stresses associated with the effects of human-driven climate change. These new varieties will not only be suitable for controlled greenhouse conditions but also for outdoor cultivation in order to maintain or even enhance the yield under adverse environmental conditions. On the other hand, the use of CRISPR/Cas could enable the design and production of new tomato varieties with better-quality fruits enriched in vitamins and other essential or bio-healthy compounds. However, in many countries, the regulatory management of crops generated using NGTs does not differ from that of crops produced using conventional genetic improvement techniques, such as mutagenesis, while in other countries, the regulation of NGT crops parallels that of GMOs. This has resulted in regulatory oversight, making it difficult to generalize scientific advances aimed at improving food security in developing countries.

## Figures and Tables

**Figure 1 ijms-25-02606-f001:**
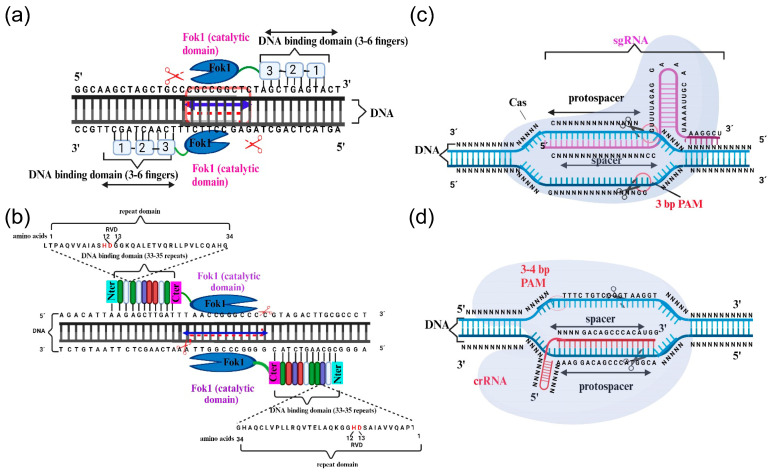
Overview of the gene-editing systems analyzed in this review. (**a**) ZFNs, (**b**) TALENs, (**c**) CRISPR/Cas9, and (**d**) CRISPR/Cas12a. The main text provides a thorough description of all the elements depicted in the different panels. Created with BioRender.com (accessed on 12 January 2024).

**Figure 2 ijms-25-02606-f002:**
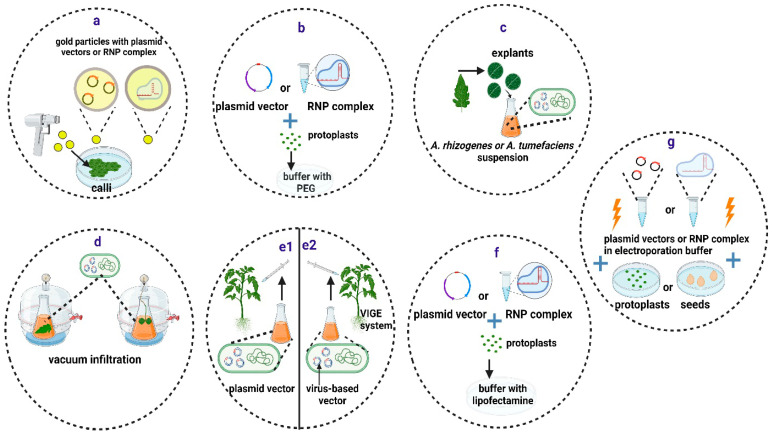
Methods for delivery of CRISPR/Cas components into plant cells and tissues. (**a**) Particle bombardment. (**b**) PEG-mediated transfection. (**c**–**e**) Biological methods, including Agrobacterium-mediated stable transformation (**c**), vacuum infiltration (**d**), and direct injection of the Agrobacterium culture with an integrative plasmid vector (**e1**) or a virus-based plasmid vector (**e2**). (**f**,**g**) Other methods, including liposome-mediated transfection (**f**) and electroporation (**g**). Created with BioRender.com (accessed on 12 January 2024).

**Figure 3 ijms-25-02606-f003:**
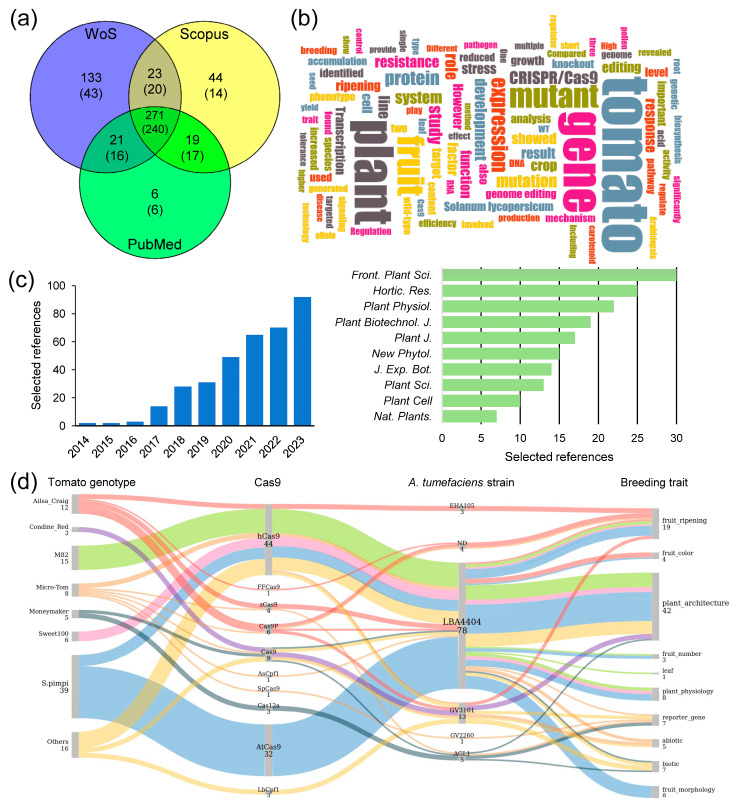
Results of a reference search and text mining analysis. (**a**) Venn diagram of scientific articles downloaded from public databases using the search term: “(Cas9 tomato) NOT (review)” and restricted by date (from 1 January 2014 to 31 December 2023). Numbers in brackets indicate selected references after manual curation. (**b**) A word cloud generated with the 356 unique abstracts selected based on the criterion of including any mutant generated by gene editing in tomato and related species in the main text. The image was generated using the web application located at https://wordclouds.ethz.ch/ (accessed on 12 January 2024). (**c**) Selected papers about gene editing in tomato and related species, along with the top 10 plant science journals in which they were published. The raw data utilized for preparing panels a–c can be found in Appendix A. (**d**) The Sankey diagram illustrates the relationship between tomato cultivars, types of Cas enzyme, *A. tumefaciens* strains, and the key breeding traits it influences. The diagram was generated with SankeyMATIC and is based on data from Appendix A. The thickness of the lines and the colors represents the proportion of evidence connecting the different categories. Spimpi: *S. pimpinellifolium* genotypes; ND: not included.

**Table 1 ijms-25-02606-t001:** Advantages and disadvantages of gene-editing systems in plants.

Editing System	Advantages	Disadvantages
ZFNs	First editing tool made available.	Low editing efficiency. High rates of off-target mutations. Assembling the ZFN array is time-consuming and requires a high level of expertise. Sensitive to DNA methylation. Not suitable for gene target multiplexing. Sequence length limitations in the target sequence.
TALENs	Targets any DNA sequence. Fewer off-target mutations. No length limitations in the target sequence.	Sensitive towards DNA methylation. Expensive and time-consuming design. Not appropriate for targeting multiple genes simultaneously.
CRISPR/Cas	Higher editing efficiency. Easier to design and relatively cheaper. Possibility of gene target multiplexing. Cas proteins work across different species. Low rates of off-target effects or no off-target effects if the sgRNA is optimized.	The choice of target gene is limited by the PAM motif.

## Data Availability

All data used for this review have been included in the Appendix A.

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
