# Peer review of "Recent Advances in Tomato Gene Editing"

_ijms, 2024, doi:10.3390/ijms25052606_

Round 1

Reviewer 1 Report

Comments and Suggestions for Authors

Comments for Authors

In this review article entitled ‘Recent advances in tomato gene editing’ authors aim to provides an overview of gene editing techniques used in plants, focusing on the cultivated tomato and its wild relatives. Authors also elaborated that several dozen genes that have been successfully edited with the CRISPR/Cas system have been selected to illustrate the possibilities of this technology to improve fruit yield and quality, tolerance to pathogens or the response to drought and soil salinity, among other factors. Authors have provided some examples of how the domestication of wild species can be accelerated using CRISPR/Cas to generate new crops better adapted to the new climatic situation or for use in indoor vertical agriculture.

I have some concerns which need to be addressed before this review article is accepted for publications.

Comments

1. The overall manuscript needs to be checked carefully for typos and grammatical errors.

2. Authors need to clarify that what are the NGTs (New Generation Technologies) mentioned in the article, and how have they contributed to crop gene editing?

3. Authors should also focus on why has the CRISPR/Cas-based editing system emerged as the preferred option among ZFNs, TALENs, and CRISPR/Cas for crop gene editing?

4. Authors need to explain that how does the high efficiency of gene editing induced by Cas proteins contribute to the appeal of the CRISPR/Cas system?

5. In what ways does the flexibility in the nucleotide sequence of RNA guides in the CRISPR/Cas system impact experimental design for genetic modification in tomatoes? Need to explain please

6. What evidence in this review article supports the utility of the CRISPR/Cas system for genetic editing in both cultivated tomatoes and related wild species?

7. Authors need to cite these two recent and relevant literature doi.org/10.1007/s10142-023-01162-5 and doi.org/10.1016/j.jplph.2019.152997

8. Authors need to elaborate that in what environments are the new varieties of tomato plants developed through CRISPR/Cas expected to be suitable, and what benefits do they offer under adverse environmental conditions?

9. Authors should also mention the differences exist in the regulatory management of crops generated using NGTs in different countries, and how does this impact the advancement of scientific improvements in food security.

10. How has the regulatory oversight of NGT crops, especially in countries where it parallels that of GMOs, affected the generalization of scientific advances aimed at improving food security in developing countries?

Thank you

Comments on the Quality of English Language

The overall manuscript needs to be checked carefully for typos and grammatical errors. 

Reviewer 2 Report

Comments and Suggestions for Authors

Dear Authors,

Thank you for the opportunity to contribute to your work as a reviewer. I hope that my constructive comments will be beneficial for your manuscript.

The topic of the manuscript is to introduce the achievements in tomato genetic modification in the last decade. The focus of the ms is dual; both methodological background and exact achievements in tomato editing is introduced in details. I feel, that the balance of the two parts is not ideal, the majority of the ms deals with the methodologies. Also, exact achievements are discussed from the perspective of the methods, while the outcomes, and also the practical benefits are less emphasized.

General comments

1.       The abstract highlights two motivations towards the justification of/need for gene editing, these are climate change and adaptability to indoor farming. The subchapter 4.3.3 deals with stresses, where the lines 522-564 deals with biotic sources, which has rather indirect connection with climate change. The lines 565-589 focuses only to climatic challenges, which seems very short in the view of its role in the abstract.

2.       Regarding indoor farming (line 463-484), I cannot see the link between the scientific results and the practice. The authors do not emphasize the importance of the results.

3.       Figure 3 contains a lot of interesting information; however, the authors do not discuss its content. Also, I was unable to find the in-text reference for this illustration.

4.       The conclusions section is rather a justification for using CRISPR/Cas, and minimal is mentioned about tomato itself. I strongly suggest rewording it.

5.       Table S1 is quite exhaustive, it is a huge and raw database, and no information is provided about its content in keywords. I am unsure, whether it might be useful for the readers. I suggest adding the meaning of red coloring (discarded articles).

Detailed comments

line 31: How do you mean highest produced vegetable? I assume, in quantity, please reword.

line 524: I suggest adding few words here to mention, that the referenced studies are systematic reviews.

Reviewer 3 Report

Comments and Suggestions for Authors

General comments

The analyzed manuscript conducts a detailed study on the latest gene editing technologies in plants, used for improving the most important productivity, quality and stress tolerance characteristics, taking as a case study one of the most important vegetable species - tomatoes

As is known, genome editing tools have the potential to change the genomic architecture of a genotype in locations and with the desired precision.

These tools have been effectively used in recent decades to identify valuable traits in plants and to generate plants with high crop yields and tolerance to biotic and abiotic stressors. The most important techniques used for genome editing in plants are: homologous recombination (HR), zinc finger nucleases (ZFN), transcription activator-like effector nucleases (TALEN), pentatricopeptide repeating proteins (PPR), CRISPR / Cas9 system, ARN interference (RNAi), cisgenesis and intragenesis.

Genome editing is therefore an invaluable tool to discover the functions of new genes, alleles and haplotypes, which can be strategically used in programs to improve crop yield and quality. Also, in the field of functional genomics, the use of gene editing techniques and the range of technological advances available to explore the gene-function relationship are of paramount importance.

The manuscript represents a courageous scientific approach, on a topic of great importance for agricultural sciences in general and horticultural sciences in particular. The analysis mode is detailed, based on comparing the information of specialized studies existing in the main flow of information. There are also presented the advantages and some drawbacks of using gene editing systems in plants, as well as the main gene editing methods. Interesting and very suggestive is also the graphic presentation that supports the text, both for the methodology stages and in the specific analyzed literature in S. lycopersicum.

The conclusions clearly show that from the multitude of technologies, for tomatoes, gene editing by applying CRISPR/Cas methodologies ensures greater precision, facilitating the obtaining tolerant genotypes  to the multitude of stressors generated by climate changes.

Congratulations and good luck!

Author Response

Please see atachment.

Round 2

Reviewer 1 Report

Comments and Suggestions for Authors

Authors have extensively improved the manuscript by addressing all the raised issues. I think the paper can be accepted for publication.